# Effectiveness of the Biography and Life Storybook for Nursing Home Residents: A Quasi-Experimental Study

**DOI:** 10.3390/ijerph19084749

**Published:** 2022-04-14

**Authors:** Doraisamy Guna, Coral Milburn-Curtis, Hui Zhang, Hongli Sam Goh

**Affiliations:** 1Nursing Administration, Sunlove Nursing Home, Singapore 534190, Singapore; 2Green Templeton College, University of Oxford, Oxford OX2 6HG, UK; coral.milburn-curtis@gtc.ox.ac.uk; 3Alice Lee Centre for Nursing Studies, National University of Singapore, 10 Medical Drive, Singapore 117597, Singapore; nurzh@nus.edu.sg (H.Z.); nurghs@gmail.com (H.S.G.)

**Keywords:** nursing home, biography, life storybook, reminiscence, long-term care, quality of life

## Abstract

There are currently limited studies that have examined the use of the biography and life storybook (BLSB) among the Asian older adult populations in the long-term care setting. This quasi-experimental study aimed to examine its impact on life satisfaction scores, depression, and quality of life among nursing home residents in Singapore. Two wards were assigned to either the intervention or control group. The intervention group was assigned to the BLSB intervention, which comprised eight nurse-facilitated structured sessions over three months and their usual daily activities, whereas the control group was assigned to the routine activities. A total of 74 nursing home residents completed the study, with 37 in each group. The BLSB intervention improved depression, quality of life, and life satisfaction for nursing home residents in Singapore, with significant results observed across all three outcomes over the 3-month period. The study findings support the use of BLSB as an effective reminiscence-based intervention for older adults in an Asian nursing home setting.

## 1. Introduction

The biography and life storybook (BLSB) involves documenting and compiling an individual’s life experiences, interests, and meaningful events, both past and present, into a single document [1]. It also comprises various artifacts—diaries, books, photographs, audiovisual recordings, personal collection items, and art pieces that help to illuminate an individual’s memorable events or experiences [2]. The BLSB creates an opportunity for people to talk about their life experiences, build connections with others, and develop a sense of identity and continuity [2]. Its use has been traced to social work and healthcare to understand foster children’s and older adults’ life stories during the late 1960s [2,3]. Since then, it has been adapted for various populations, such as older adults and people with intellectual disabilities.

Although various approaches and formats guide BLSB interventions, Kindell et al. identified several key features of BLSB implementation for older adults [3]. The first step requires trained staff to get participants to share their life stories and collate artifacts. This process helps establish “new connections” by encouraging meaningful conversations between staff, patients, and family members. At the same time, it allows “interactional connections” [3]. The second step involves the production of the BLSB, which supports the recall of their life stories and creates a shared knowledge about their “identity”. There are several resources and guides on how to produce a BLSB for the older adult population, such as Dementia U.K.’s Life Story Book template and the Agency for Integrated Care’s (AIC) Living with Dementia resource kit [3,4]. The final step outlines ways that the BLSB can be used. For example, the BLSB can create opportunities for older adults to re-establish or enhance “emotional connections” with their families. Another advantage is its “practical care connections”, whereby the BLSB allows care staff to understand the individual better and formulate a more appropriate and person-centered care plan [3]. The concept of person-centered care has been widely advocated in residential care settings as it espouses the incorporation of an individual’s values and preferences to guide care plans and achieve realistic health and life goals [5]. The BLSB can promote person-centered care by allowing individuals to share memories, reconcile with their past, and achieve a sense of self-worth and positive aging.

Several systematic reviews have reported growing studies that examined the use of BLSB for patients with dementia and older adults within the long-term care setting [2,3,6,7,8]. Most studies have stated the potential benefits of BLSB in enhancing older adults’ lives, but most of the studies were qualitative. In a more recent systematic review by Doran et al. [6], they retrieved one randomized controlled trial, one quasi-experimental, and two mixed-method studies from 12 studies. Although these studies reported benefits, such as improvement in staff’s care attitudes and knowledge following BLSB intervention, and improvement in residents’ aggressive behaviors and quality of life scores (QoL), these studies were observed to have low sample sizes from 5 to 73, and were mainly conducted in Western countries. There is currently a lack of research on the Asian older adult populations in the long-term care setting. Additionally, Kindell et al. highlighted insufficient evidence supporting its use for older adults with minimal cognitive impairments [3]. Most studies were conducted on patients with moderate to severe stages of dementia. Although the authors believed that BLSB can potentially benefit the nursing home residents in Singapore, it is necessary to evaluate its effectiveness before full-scale implementation due to the heterogeneity in BLSB designs and the concerns that the interventions may trigger negative memories for individuals. This study aimed to employ a quantitative method to evaluate the effectiveness and feasibility of BLSB in Singapore residential aged care settings.

## 2. Materials and Methods

### 2.1. Study Design

The study aimed to investigate the feasibility of the BLSB intervention and examine the possible effects on life satisfaction scores, depression, and quality of life among nursing home residents in Singapore. The research questions were as follows:Is there a difference in nursing home residents’ depression scores after receiving the BLSB intervention compared to those who participated in usual daily care?Is there a difference in nursing home residents’ life satisfaction scores after receiving the BLSB intervention compared to those who participated in usual daily care?Is there a difference in nursing home residents’ QoL scores after receiving the BLSB intervention compared to those who participated in usual daily care?

This study employed a quasi-experimental design and was conducted over six months at a charitable nursing home in Singapore in 2018 [9]. As controlling for blinding and treatment contamination was impossible, we assigned residents in one ward at the nursing home as the intervention group (n = 37) and another as the control group (n = 37). Assessment of participants’ sociodemographic and outcome variables was determined at baseline before the intervention to rule out any between-group differences in extraneous factors.

### 2.2. Settings and Participants

The study was conducted in two wards at a charitable nursing home, which received government subsidies and funding. The nursing home is a 242-bed residential geriatric and dementia care facility. The nursing home admits residents who require long-term nursing care, with 50–55% of them requiring moderate functional assistance. The remaining require a high level of nursing care and functional assistance. The study’s inclusion criteria included those: (1) aged 60 years or above; (2) having normal or mild cognitive impairment with a cut-off Mini-Mental State Examination (MMSE) score of 19 or more; and (3) able to communicate and participate in the study. Exclusion criteria included individuals: (1) having a medical condition that results in moderate to severe impairment; (2) having a hearing or speech impairment; or (3) diagnosed with severe depression for welfare and safety reasons.

### 2.3. Intervention

The BLSB intervention was developed from our literature review and Singapore’s Agency for Integrated Care Dementia Resource Kit. It comprised eight structured sessions over three months (four weekly sessions for the first month, followed by another four bi-weekly follow-up sessions for the next two months) in addition to the participants’ usual daily activities. The BLSB protocol comprised either nurse-facilitated one-to-one or group sessions lasting between 45 and 90 min. The first four sessions involved autobiography, photo reminiscence, and artifact collections from a caregiver. Each participant’s life story was compiled into an individual life storybook at the end of these sessions. The participants were strongly encouraged to actively participate in the production by providing photos, giving feedback, and choosing the design of the life storybook (Figure 1—BLSB Protocol). The subsequent four sessions would then involve them conducting sharing sessions with fellow residents, family members, and friends. The intervention group also participated in their usual daily activities in addition to the BLSB intervention. The control group only participated in their routine daily activities, including physical exercises, recreational activities, and self-care training.

### 2.4. Data Collection

The study questionnaire comprised four parts: the participants’ demographic data, emotional status, life satisfaction, and quality of life. Demographic information was collected at baseline and included the following information: age, gender, ethnicity, religious beliefs, marital status, educational level, and financial support. Three validated assessment instruments were used to collect the variables of interest for both intervention and control groups at specified time points. Emotional status was measured using the Geriatric Depression Scale-15 (GDS-15), a widely used 15-item instrument for assessing depression in older adults in various settings, including nursing home settings. The instrument has a reported pooled sensitivity of 86% and 76% [10] and has been well-validated for the local setting [11]. Each item is a close-ended question with either a “Yes” or “No” response on how the participant agreed to statements reflecting the psychophysiological aspects of depression for the previous week. Higher scores reflect depression, with a cut-off of 10 or above suggestive of a major depressive episode [10]. Life satisfaction was measured using a modified 20-item Life Satisfaction Index (LSIA). It reflected the participant’s subjective well-being, such as mood tone, zest versus apathy, congruence between desired and achieved goals, and self-concept. Each item requires a response of “agree”, “disagree”, or “don’t know”, and a point was given for “agree”, and no point for “disagree” or “don’t know”. The total score ranges from 0 to 20, with higher scores reflecting higher satisfaction. The LSIA has been validated in similar studies on the older adult population [12,13,14]. Quality of life was measured using a modified 50-item QoL scale for Nursing Home Residents (QoL-NHR), which examines 11 QoL domains, such as functional competence [15]. Each item consists of a 4-point Likert scale ranging from “never—1” to “often—4”. The total score ranges from 50 to 200, with higher scores reflecting the residents’ better-reported quality of life.

The questionnaire was used to collect data at five different time points during the 3-month study period—Week 1, Week 2, Week 4, Week 8, and Week 12. All recruited participants were asked to complete the questionnaire before the study commencement in Week 1. In addition, participants in the intervention group were asked by the researcher (GD) to complete the same questionnaire following the BLSB sessions. In contrast, those in the control group completed the questionnaire at the end of their usual daily activities at the nursing home on a selected day.

### 2.5. Data Analysis

Data were stored electronically in a secure manner prescribed by university guidelines and analyzed using IBM SPSS Version 24.0. Descriptive statistics were used for univariate analysis to present sociodemographic characteristics, and inferential statistics, independent *t*-test, or ANOVA were used to examine between-group differences at baseline. In addition, repeated-measures ANOVA was employed to assess treatment effects on intervention and control groups over the different time points. The significance level was set at 0.05 for hypothesis testing.

## 3. Results

A total of 74 nursing home residents completed the study, with 37 in each group. Their demographic data are shown in Table 1. The mean age was 73.0 for the intervention group (SD = 9.16) and 73.2 (SD = 8.37) for the control group. No significant difference was observed between the demographic characteristics except for ethnicity.

### 3.1. GDS-15 Depression Scores

The mean GDS-15 depression scores by group over time are shown in Table 2. Before the intervention, no significant difference was observed in the GDS-15 scores between both groups. However, the average depression score in the intervention group decreased from 8.4 in Week 1 (baseline) to 4.2 at the end of Week 12. In contrast, the average depression score in the control group remained relatively unchanged (Table 2). Using Pillai’s Trace, there was a significant effect of the intervention on the depression score for the nursing home residents in the intervention group (V = 0.61, F (5, 68) = 21.57, *p* < 0.01). Based on difference-in-difference estimations, significant differences in outcome were observed from Week 4 onwards.

### 3.2. LSIA (Life Satisfaction) Scores

The mean LSIA scores by group over time are shown in Table 3. We observed a significant difference in the LSIA scores between both groups at the baseline. Nevertheless, the average life satisfaction score in the intervention group increased from 10.2 in Week 1 (baseline) to 12.3 at the end of Week 12. In contrast, the average LSIA scores in the control group remained relatively unchanged (Table 3). Using Pillai’s Trace, there was a significant effect of the intervention on the LSIA scores for the nursing home residents in the intervention group (V = 0.29, F (5, 68) = 5.62, *p* < 0.01). Based on difference-in-difference estimations, significant differences in outcome were observed at Week 12.

### 3.3. QoL (Quality of Life) Scores

The mean QoL scores by group over time are shown in Table 4. Before the intervention, no significant difference was observed between both groups’ baseline QoL scores. However, the average QoL scores in the intervention group increased from 144.6 in Week 1 (baseline) to 155.7 at the end of Week 12. In contrast, the average QoL scores in the control group remained relatively unchanged (Table 4). Using Pillai’s Trace, there was a significant effect of the intervention on the QoL scores for the nursing home residents in the intervention group (V = 0.80, F (5, 68) = 53.2, *p* < 0.01). Based on difference-in-difference estimations, significant differences in outcome were observed from Week 1 onwards.

## 4. Discussion

The BLSB intervention effectively improved depression, quality of life, and life satisfaction for nursing home residents in Singapore. Our analysis showed that the 12-week BLSB intervention resulted in a significant decrease in depression scores among the intervention group compared to the control group based on the GDS-15 depression scores. This result was consistent with several systematic reviews that reported a small to medium effect of BLSB in reducing depressive symptoms among the general older adult populations [8,16,17]. Depression has been considered a major health risk among the older adult population due to their increasing vulnerabilities to physical and mental decline. Institutionalized nursing home residents are more likely to experience depression than the general population, as many of them experienced multiple health issues and lacked social contact and autonomy [18,19,20]. Although aging is an inevitable part of life, depression need not be part of it. Early recognition, diagnosis, and treatment can counteract and prevent depression’s emotional and physical consequences. BLSB can significantly mitigate social loneliness and depression among the older adult population [17].

Our study also found that BLSB effectively promoted life satisfaction among nursing home residents. This finding was consistent with two reviews that reported a significant pooled effect favoring the reminiscence-based intervention over control [16,17]. In addition, Tam et al. reported no differences in the BLSB effect on life satisfaction between the type of sessions (individual versus group) [17]. We also found that the BLSB can potentially improve the quality of life for nursing home residents, which concurs with the results of other similar controlled trials [21,22,23]. However, our result differs from the meta-analysis by Tam et al., who found no significant effect of BLSB on quality of life [17]. One possible reason for the significance of our results may be the time and effort spent to assist the subjects in recalling positive memories and integrating them into a personal collection. Through the process of recall, collection, and sharing of one’s memories and artifacts, the subjects were given greater opportunities to develop “connections” with staff and other residents, thus improving their sense of self-worth and perceived quality of life [16,23].

The mechanism by which BLSB demonstrated benefits for our subjects can be attributed to several important factors. The first factor involved a theoretical framework that guided the design of the life storybook. The BLSB design created opportunities for the residents in the intervention group to reflect on and share the various stages of their lives. This process provided positive affirmation of their lives and helped them establish a sense of identity after months or years of “de-personalization” at the nursing home [3,24]. The second factor pertains to the BSLB structure. Our BLSB intervention comprised eight structured group sessions over three months. Each session was kept to approximately 60 min and a maximum number of 5–6 participants to control the quality of the sessions. The sessions were facilitated by a trained nurse and incorporated as part of the participants’ usual daily activities. The intervention provided the opportunity for “interactional connections” and “establishing of new connections” among the participating residents [3]. This design was consistent with the recommendations by Yen and Lin, who conducted a systematic review of 16 studies and reported the effective elements underlying successful reminiscence therapy in Taiwan [25]. The third factor pertains to the outcome of the BLSB, with the formation of a life storybook shared with family members, fellow residents, and staff. The life storybook served as a platform for residents to establish “emotional connections” with their families [3]. It also established “practical care connections” for staff as they recognized the residents as individuals with their own particular needs and preferences.

In summary, the positive finding supported the use of BLSB as an effective reminiscence-based intervention for older adults in an Asian nursing home setting. Globally, there has been increasing focus on promoting person-centered care across continuing care settings, emphasizing the need to enhance their quality of life and life satisfaction. Our study confirmed the benefits of BLSB not just for people living with dementia, but also for the general older adult population [5]. It is essential to continue understanding what successful aging looks like in older individuals to adopt meaningful practices and interventions to elicit successful aging responses in those living in the communities. In addition, there is a need to pay more attention to depressed older adults’ social aspects and a need to enhance social networks, social support, and participation in recreational and leisure activities to improve older adults’ life satisfaction and QoL in the social domain. The study has added to the limited Asian studies that have examined the feasibility of BLSB among the older adult population in a nursing home setting [8,24].

There were several limitations to this study. First, the study was conducted on a single site, making it difficult to determine if the same inferences could be drawn if the study was replicated for other nursing homes. Nevertheless, this study can serve as a basis for a more extensive study involving several nursing homes in the future. Second, the Hawthorne effect may have influenced the study’s findings, whereby the participants’ reactions and behaviors became more positive in response to their awareness of being observed as part of the study [26]. As this effect could potentially make the results appear more favorable, we sought to mitigate it by using a different researcher to collect the responses from the participants [26]. Third, the sample size and lack of randomized participant allocation might have limited the study’s statistical power and contributed to the differences in ethnicity allocation and life satisfaction scores at baseline [9]. Although our study observed some significant differences in ethnicity and life satisfaction scores at baseline, our difference-in-difference estimations showed that all three outcomes were significant due to treatment effects. The risk of treatment contamination was also ensured as participants in both groups usually only mingled within their own residential areas, with minimal interactions between groups due to the facility layout [9]. Moving forward, future studies with a larger sample size are needed to validate its benefits in these settings, and to determine its impact on staff at long-term care facilities.

## 5. Conclusions

Although successful aging was not a clear topic of analysis in the biomedical literature until the early 1960s, there have been several endeavors to comprehend how to promote longevity and positive states of health once people start becoming older. Aging is a complex process, with contemporary psychiatrists and psychologists thinking that later life may result from initial development tasks, or a period of consistent growth and disagreements that need to be discussed. Hence, the BLSB is particularly suitable for older adults who face a loss of meaning in life and hold a negative view of themselves. If appropriately implemented, BLSB can help provide better care of the aging population, as it can help reduce depression and enhance the quality of life and life satisfaction.

## Figures and Tables

**Figure 1 ijerph-19-04749-f001:**
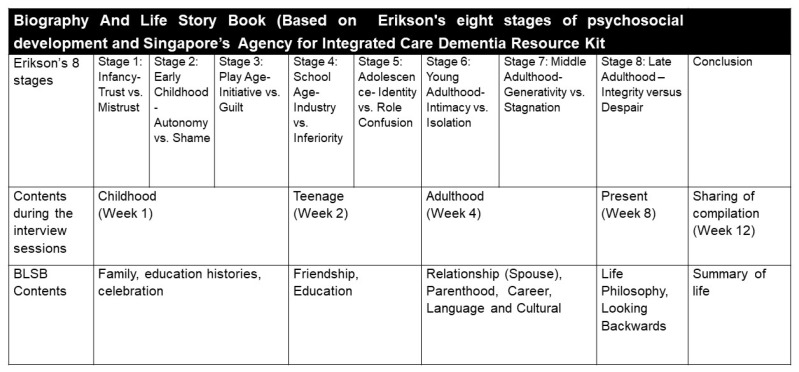
Design of biography and life storybook protocol.

**Table 1 ijerph-19-04749-t001:** Demographic characteristics and baseline outcomes of participants by group.

Demographic Characteristic	Intervention (n = 37)		Control (n = 37)		χ^2^	*p*-Value
	n	(% Within Row)	n	(% Within Row)		
Gender					1.51	N.S.
Male	22	(44.9)	27	(55.1)		
Female	15	(60.0)	10	(40.0)		
Marital Status					4.96	N.S.
Married	23	(52.3)	21	(47.7)		
Single	8	(72.7)	3	(27.3)		
Divorced/Separated	3	(33.3)	6	(66.7)		
Widowed	3	(30.0)	7	(70.0)		

Ethnicity					12.92	<0.05
Chinese	13	(31.7)	28	(68.3)		
Malay	13	(76.5)	4	(23.5)		
Indian	10	(66.7)	5	(33.3)		
Others	1	(100)	0	(0)		

Religious beliefs					2.66	N.S.
Christianity/Catholicism	6	(28.6)	15	(71.4)		
Buddhism	8	(40.0)	12	(60.0)		
Taoism	0	(0)	1	(100)		
Islam	13	(65.0)	7	(35.0)		
Hinduism	9	(81.8)	2	(18.2)		
Others	1	(100)	0	(0)		

Educational level					2.28	N.S.
Primary school and below	27	(51.9)	25	(48.1)		
Secondary level	9	(45.0)	11	(55.0)		
Tertiary level	1	(50.0)	1	(50.0)		

	M	SD	M	SD	t	*p*-value
GDS-15 scores (Baseline)	8.43	1.04	8.73	1.24	1.12	0.268

LSIA scores (Baseline)	10.24	1.40	9.00	1.78	−3.34	<0.05

QoL-NHR scores (Baseline)	144.64	2.62	146.72	4.55	2.41	0.946

The mean difference is significant at the 0.05 level. N.S.: Not significant. DID—difference-in-difference estimation; t—independent *t*-test; M—mean; SD—standard deviation; GDS—Geriatric Depression Scale-15; LSIA—Life Satisfaction Index; Quality of Life scale for Nursing Homes Residents (QoL-NHR).

**Table 2 ijerph-19-04749-t002:** Comparison of mean GDS-15 scores between groups over time.

**GDS-15 Scores**	**Intervention** **(n = 37)**		**Control** **(n = 37)**		**DID (Sig.)**	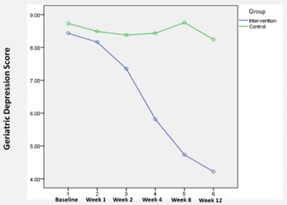
	**Mean**	**(S.D.)**	**Mean**	**(S.D.)**		
Baseline (T1)	8.43	1.04	8.43	1.04	-	
Week 1 (T2)	8.16	1.50	8.49	1.43	−0.027	
Week 2 (T3)	7.35	1.51	8.38	1.36	−0.730	
Week 4 (T4)	5.81	0.91	8.43	1.34	−2.324 *	
Week 8 (T5)	4.73	1.17	8.76	1.36	−3.730 *	
Week 12 (T6)	4.22	0.95	8.24	1.04	−3.730 *	

* The mean difference is significant at the 0.05 level; DID—difference-in-difference estimation.

**Table 3 ijerph-19-04749-t003:** Comparison of mean LSIA scores between groups over time.

**LSIA Scores**	**Intervention** **(n = 37)**		**Control** **(n = 37)**		**DID (Sig.)**	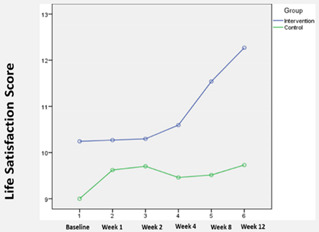
	**Mean**	**(S.D.)**	**Mean**	**(S.D.)**		
Baseline (T1)	10.24	1.40	9.00	1.78	-	
Week 1 (T2)	10.27	1.22	9.62	1.23	−0.595	
Week 2 (T3)	10.30	1.13	9.70	1.22	−0.649	
Week 4 (T4)	10.59	1.57	9.46	1.04	−0.108	
Week 8 (T5)	11.54	2.60	9.51	1.24	0.784	
Week 12 (T6)	12.27	2.13	9.73	1.10	1.297 *	

* The mean difference is significant at the 0.05 level; DID—difference-in-difference estimation.

**Table 4 ijerph-19-04749-t004:** Comparison of mean QoL scores between groups over time.

**QoL-NHR Scores**	**Intervention** **(n = 37)**		**Control** **(n = 37)**		**DID (Sig.)**	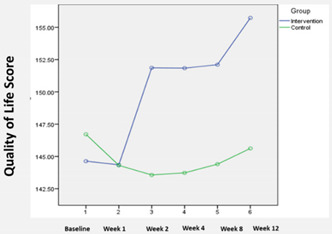
	**Mean**	**(S.D.)**	**Mean**	**(S.D.)**		
Baseline (T1)	144.64	2.62	146.72	4.55	-	
Week 1 (T2)	144.35	2.58	144.31	2.58	2.122 *	
Week 2 (T3)	151.86	4.11	143.57	1.94	10.378 *	
Week 4 (T4)	151.84	3.19	143.73	1.92	10.189 *	
Week 8 (T5)	152.11	3.28	144.41	2.54	9.784 *	
Week 12 (T6)	155.73	3.05	145.62	2.63	12.189 *	

* The mean difference is significant at the 0.05 level; DID—difference-in-difference estimation.

## Data Availability

The data used and/or analyzed during the current study are available from the corresponding author on request.

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
