# Peer review of "Effectiveness of the Biography and Life Storybook for Nursing Home Residents: A Quasi-Experimental Study"

_ijerph, 2022, doi:10.3390/ijerph19084749_

Round 1
Reviewer 1 Report
The subject of the study is very interesting and its results can contribute to the development of similar interventions for the elderly in long-term care institutions.
Some suggestions for improving the article:
Part 3.3 Lines 176-182 are supposed to explain the findings relating to quality of life. It seems that data relating to life satisfaction were erroneously provided.
2. In the discussion, lines 234-236, the issue of elderly-centered care is first mentioned, I suggest adding a brief reference to this in the introduction section as well.
3. In the discussion in line 247 the Hawthorne effect is mentioned without references and without a brief explanation for those who do not know what it is. Suggests adding a short explanation.
Author Response
Dear Reviewer 1,
The attached cover page and revised manuscript

Reviewer 2 Report
I think it is a very interesting article. It is well-written and structured. The authors do a good job setting the motivation and justifying the study.
My main comments are:
- You could add a difference-in-difference indicator to your results, just to make the case that the intervention worked, particularly when you couldn't ensure similar outcomes at baseline.
- Limitations can be improved to better understand how the study design addressed some issues and how the results should be interpreted
- I was not hear of this strategy and found it fascinating. Results presented are encouraging and—unlike other articles—I would like to see what other extensions in this line of research are possible. For example, how this intervention impacts not only residents but staff in long-term care facilities.

Author Response
Dear reviewer, Thank you so much for your comments. We have made the revisions as suggested. Please refer to the attached documents.
